# Four-Year Trajectories of Health-Related Quality of Life in People Living with HIV: Impact of Unmet Basic Needs across Age Groups in Positive Spaces, Healthy Places

**DOI:** 10.3390/ijerph182212256

**Published:** 2021-11-22

**Authors:** Phan Sok, Mary V. Seeman, Rosane Nisenbaum, James Watson, Sean B. Rourke

**Affiliations:** 1Institute of Medical Science, University of Toronto, Toronto, ON M5S 1A8, Canada; phan.sok@gmail.com (P.S.); mary.seeman@utoronto.ca (M.V.S.); 2Department of Psychiatry, University of Toronto, Toronto, ON M5S 1A1, Canada; 3MAP Centre for Urban Health Solutions, Li Ka Shing Knowledge Institute, St. Michael’s Hospital, Toronto, ON M5B 1W8, Canada; rosane.nisenbaum@unityhealth.to (R.N.); james.watson@unityhealth.to (J.W.); 4Applied Health Research Centre, St. Michael’s Hospital, Toronto, ON M5G 1B1, Canada; 5Division of Biostatistics, Dalla Lana School of Public Health, University of Toronto, Toronto, ON M5T 3M7, Canada

**Keywords:** HIV, aging, unmet basic needs, health-related quality of life, individual growth curve, longitudinal study

## Abstract

Despite significant advances in antiretroviral therapy, unmet basic needs can negatively impact health-related quality of life (HRQoL) in people living with HIV, especially as they age. We aimed to examine the effect of unmet basic needs across age groups on changes in HRQoL over a 4-year period in persons with HIV. Physical and mental HRQoL scores from the Positive Spaces, Healthy Spaces cohort interviewed in 2006 (n = 538), 2007 (n = 506), and 2009 (n = 406) were examined across three age groups according to their unmet needs for food, clothing, and housing. Individual growth curve model analyses were used to investigate changes over time, adjusting for demographics, employment, living conditions, social supports, HIV status, and health behavior risks. Low scores on physical and mental HRQoL were positively associated with higher number of unmet basic needs (β = −6.40, standard error (SE) = 0.87, *p* < 0.001 and β = −7.39, SE = 1.00, *p* < 0.001, respectively). There was a slight improvement in physical and mental HRQoL over 4 years in this HIV cohort, but the burden of unmet basic needs took its toll on those over 50 years of age. Regularly assessing unmet basic needs is recommended given the impact these can have on HRQOL for people living with HIV. Recognition of unmet needs is vital, as is the development of timely interventions.

## 1. Introduction

Currently, an estimated 37.7 million people are living with HIV globally, 27.5 million of whom, at the end of 2020, were on antiretroviral therapy [1]. In 2016, an estimated 5.7 million were older adults (aged 50 or over) [2]. In Canada, the most recent available data indicate that 88,881 individuals have been diagnosed with HIV since 1985 [3]. An estimated one in six of all Canadians living with HIV (15 years or older) will reach the age of 50 years within the next five years [4].

People living with HIV face numerous health challenges. HIV infection can lead to persistent inflammation [5], as well as chronic immune activation [6] and immunosenescence [7,8], all of which accelerate the natural process of aging [9]. Aging itself is frequently accompanied by additional and multiple chronic health conditions [10,11]. An accumulation of pathologies, coupled with the natural process of aging, too often negatively affects the physical health [12,13], mental health [14,15], and social function of persons with HIV [16], resulting in early and excessive frailty [17,18].

In addition, a variety of the social determinants of health impinge on older persons living with HIV and can compromise their health-related quality of life (HRQoL). Important factors are poverty [15], unemployment [19], food insecurity [20], housing needs [21], isolation and loneliness [22,23], lack of social support [24], social and institutional stigma [19,23,24], depression [25], and substance abuse [25,26], all of which interact with and increase the severity of age-related illnesses [25,27,28,29,30,31].

The concept and definition of unmet basic needs vary. Its exact scope depends on the circumstances of the target population and, with reference to the HIV population, can include: gaps in social services or inaccessibility of drug treatment, or lack of medical care for women or for those with concurrent addiction [31]. It can depend on access to specific foods or hygiene products, or on adequate shelter for homeless women [32]. Some equate it with inadequate food, housing, transportation services, and drug use [33], some with challenges in accommodation, food security, benefits, and money management, as well as mental illness [34]. Some specify deficiencies in food and shelter, plus a subjective sense of insecurity, as well as viral hepatitis infection [35]. 

In a previous study by our group [36], we defined unmet basic needs as the inability to pay for food, clothing, housing-related costs (i.e., rent, mortgage, property tax, or utilities), or being at subjective risk for homelessness. The findings of that study were that both physical and mental HRQoL were negatively associated with the presence of any unmet basic need, a highly prevalent (87%) condition in our sample. With respect to HRQoL, there have been very few publications addressing this outcome over time in persons living with HIV [37]. 

HRQoL is becoming increasingly relevant because, despite lifespan improvement of persons with HIV since the advent of antiretroviral therapy [38], adults with HIV continue to suffer from premature aging, as described above [39]. Furthermore, poor quality of life intersects with social determinants of health and can also lead to early mortality [40]. 

The focus of the present study is to investigate the changes of physical and mental HRQoL over time, and to determine whether these changes vary with the number of unmet basic needs at progressive ages. We hypothesized that both physical and mental HRQoL would change significantly over time and that the burden of unmet basic needs would be negatively associated with HRQoL over the period of the study. We also hypothesized that these associations would be most pronounced in the oldest study participants. 

## 2. Materials and Methods

### 2.1. Positive Spaces, Healthy Places (PSHP) Cohort

The PSHP cohort was a five-year observational study that evaluated the effects of housing and other supports on the health and well-being of a community-based sample of adults living with HIV who came from key populations in the province of Ontario, Canada. A detailed description of the PSHP cohort is available elsewhere [41]. In summary, to be eligible for the cohort, participants had to be HIV positive, 18 years of age or older, able to provide informed consent, and resident in the Canadian province of Ontario. All data were initially collected via face-to-face interviews conducted by trained peer research associate (PRA) interviewers who, themselves, were persons living with HIV. Each participant received a CAD 40 honorarium at each interview. 

### 2.2. Sample and Data Analyses

Peer research assistants interviewed 602 participants in 2006, re-interviewed them in 2007 (n = 509), and again in 2009 (n = 442). Eligibility for the current study was participation in at least two of these interviews. This resulted in a total of 1486 observations from 538 participants in 2006 (baseline), 506 participants in 2007, and 442 participants in 2009. Those who are known to have died between 2007 and 2009 (n = 18) or those who died in 2009 (n = 2) were included in the analyses, but those who died between 2006 and 2007 (n = 15) were excluded. The physical and mental HRQoL scores for those who died between 2007 and 2009 were, as would be expected, significantly lower relative to those who survived. We note that initial physical and mental HRQoL scores of those who were not available or lost for follow-up interviews in 2007 or 2009 were not statistically different from those who remained in the cohort over all the years of the study. Compared to those who attended all three interviews, there were no significant differences at baseline in the number of unmet basic needs in those who missed a subsequent follow-up interview. However, those who missed their follow-up appointment in 2007 were significantly more likely than others to report significant alcohol or drug use, history of incarceration or homelessness, and depression at their baseline interview. Those who were missing the 2009 interview were significantly more likely than others to be male, gay, bisexual, or be men who have sex with men, middle-aged, living alone, and financially disadvantaged. 

### 2.3. Measures

Conceptualizing basic needs. For this study, we defined ‘unmet basic needs’ as the inability to pay for food (Q. 60a), clothing (Q. 61a), housing-related costs (i.e., rent, mortgage, property tax or utilities, Q. 23), and, also, those who endorsed being at risk for homelessness (Q. 57) [36]. The internal consistency of the four items over time (i.e., food, clothing, housing-cost needs, and risk of homelessness) yielded a four-year Cronbach’s alpha of 0.60. 

Outcome variables. The concept of HRQoL is here defined as a person’s measurable rating of his or her physical and mental health, and social functioning. These are represented by physical and mental HRQoL scores, derived from the 35-item Medical Outcomes Study-HIV Health Survey (MOS-HIV), and used previously to evaluate HRQoL [42]. Results of prior studies show that the MOS-HIV is a reliable and valid tool for measuring HIV patient functioning and well-being [43,44]. The MOS-HIV includes 10 dimensions of health and is the same instrument we used in our previous study [36]. The internal consistency of MOS-HIV in our sample yielded a four-year Cronbach’s alpha of 0.90.

Predictor. A total number of unmet basic needs was generated based on a questionnaire endorsement of items about the ability to pay for food, clothing, housing-related costs, or feeling subjectively at risk of homelessness. Unmet basic needs were calculated at each of the three survey times (2006, 2007, and 2009) and for the four-year study period.

Because the PSHP cohort was surveyed three times between March 2006 and December 2009, *time* was coded as ‘0’ (2006), ‘1’ (2007), and ‘3’ (2009) in our analyses. 

Covariates. Participant age was categorized as ‘young’ (20–34 years old), ‘middle-aged’ (35–49 years of age), and ‘old’ (50 years and older). Sixteen covariates, some time invariant and some time variant, and all known from previous studies to negatively affect the health and well-being of people living with HIV, were coded 0 and 1. Time-invariant variables were: baseline demographics such as gender (female vs. male), ethnicity (Caucasian vs. others), sexual orientation (heterosexual vs. other), and residence in the Greater Toronto area (yes vs. no). Time-variant variables were: educational level (above vs. below high school degree), employment status (employed vs. unemployed (including retired or disabled)), personal gross income (above vs. below the sample median cut-off of CAD 1150 dollars per month), living arrangements (living with someone else vs. alone), social support (above vs. below the sample median cut-off of 45 points, using the 19-item Medical Outcomes Study Social Support Survey) [45], a history of AIDS diagnosis (absent vs. present), CD4 T-cell count (above vs. below 200 cells/mL), depression (absent vs. present, using the cut-off of 20 points on the Center for Epidemiological Studies of Depression (CES-D) scales [46], drug abuse (no vs. yes, using the cut-off of six points on the Drug Abuse Screening Test (DAST) scales,) [47], alcohol use disorder (no vs. yes, using the cut-off of eight points on the Alcohol Use Disorders Identification Test (AUDIT) scales) [48], incarceration during the last year (no vs. yes) and homelessness during the last year (no vs. yes). 

### 2.4. Statistical Analysis

Descriptive analyses were performed with Chi-square (χ^2^), Cochran–Armitage Trend, Student *t*, and Kruskal–Wallis tests, as appropriate. The four-year trajectories of physical and mental HRQoL in persons living with HIV in the PSHP cohort were evaluated. 

A series of individual growth curve models [49] with mixed effects and a random effect of intercepts and slopes (*time*) were utilized to evaluate changes in physical and mental HRQoL scores over time and to determine the influence of unmet basic needs on these changes across the age groups. The mixed-effects regression method can handle missing data and irregular time measures for both time-invariant and time-varying variables [50]. For this reason, there was no necessity to correct for missing data. 

We report on six separate models of physical and mental HRQoL outcomes. In Model 1, the random intercepts and no variables were included. It serves as the baseline and estimates the initial mean of summary HRQoL scores [51]. In Model 2, *time* with random slopes was added to examine the rates of change in physical and mental HRQoL scores. In Model 3, we added unmet basic needs to assess the relationships between unmet basic needs and the two outcomes, using met needs for the reference group. In Model 4, age groups were added to evaluate the association of age on the two summary scores, using the youngest age group as the reference group. In Model 5, covariates were added while maintaining the significance level of intercepts and linear slopes [51], as well as *time*. Finally, in Model 6, we included the interactions between unmet basic needs and age groups. 

To evaluate the fitness of each model, a -2 Res Log-Likelihood (-2 LL), Akaike Information Criterion (AIC), and Bayesian Information Criterion (BIC) [51,52] were assessed for the best model fit across the six models. For the final model of each outcome, residual covariance structures (i.e., unstructured (UN), Toeplitz (TOEP), compound summary (CS), and autoregression (AR)) were assessed for the best fit model. A two-sided value of *p* < 0.05 was used to define statistical significance throughout. All data analyses were performed using the SAS^®^ OnDemand for Academics, SAS Institute Inc., Cary, NC, USA.

## 3. Results 

### 3.1. Sample Characteristics 

At baseline (2006), the median age of participants was 43.0 (IQR = 38.0, 49.0), and median education was 6.1 (4.0, 8.0). The sample consisted of 134 (24.9%) and 404 (75.1%) female and male participants, respectively. Overall, there were 54.7% men who have sex with men and 8.5% bisexual or lesbian individuals. Approximately, 11.9% of people identified as being from Africa, the Caribbean, or Black, 12.5% identified as being Indigenous or First Nations, and 2.4% identified as having Asian background. In this cohort, 19.2% were people who used drugs. Roughly 40.0% were from the Greater Toronto area. The proportion of unemployed was 79.0% for the sample. The duration of HIV infection (self-reported by participants) was 11.0 years (6.0, 16.0), and the duration of receiving combination antiretroviral therapy was 8.0 years (3.0, 11.0). A majority of participants were in the middle-aged group (60.0%), with 14.8% in the ‘younger’ and 25.2% in the ‘older’ age groups (Table 1). 

### 3.2. Unmet Basic Needs

The proportions and number of unmet needs for each period and over time are presented in Figure 1. At the 2006 interview, participants who had 3- or 4-unmet needs were 21.3% or 17.0%, respectively, and only 14.7% had no unmet (met) needs. At the 2007 interview, participants who had three or four unmet needs were 19.6% or 12.6%, respectively, and 20.0% had met needs. At the 2009 interview, participants who had three or four unmet needs were 19.9% or 15.1%, respectively, and 21.0% had met needs. There were statistically significant differences in the numbers of unmet basic needs in the 2006 and 2007 interviews, but not in the 2009 interview. The oldest participants reported more ‘met’ needs and, gradually, fewer unmet needs, while the two younger participant groups (age 20–34 years old and 35–49 years old) reported fewer ‘met’ needs over time (Table 1). 

### 3.3. Physical and Mental Health-Related Quality of Life 

At baseline (2006), the mean physical HRQoL of participants was 42.8 (standard error (SE), 0.47) and mental HRQoL was 43.9 (0.51). Both physical and mental HRQoL scores for participants were significantly decreased in parallel with the rising number of unmet basic needs (Figure 2a,b; Table 1).

### 3.4. Individual Growth Curve Models in Physical and Mental Health-Related Quality of Life

Table 2 and Table 3 display six individual growth curve models of physical and mental HRQoL. In Model 1, the initial physical and mental HRQoL scores of the study participants were below average, 43.3 (0.40) and 45.1 (0.42), respectively. When the dimension of *time* was added to Model 2, a linear increase in physical and mental HRQoL scores was observed, β = 0.51 (0.15), β = 0.78 (0.18), respectively. After adding the quadric *time*, the association was no longer statistically significant and eliminated the influence of *time*. For this reason, the quadric *time* was not kept in this model nor in the following models. When unmet basic needs were separately added to Model 3, we observed that increasing low scores of physical and mental HRQoL were significantly associated with numbers of unmet basic needs. The interactions between unmet basic needs and *time* were not significant, suggesting that unmet basic needs remained relatively the same over time. After adding age groups into Model 4, unmet basic needs remained significant (Figure 2), while only physical HRQoL scores were significantly influenced by age. In Model 5, covariates were introduced. Both numbers of unmet basic needs and age groups remained with no changes in their significant levels. Covariates, such as residence outside Greater Toronto, being Caucasian, having low social support, and using drugs were significantly and negatively associated with physical HRQoL scores. Being female, unemployed, having low income, using substances, and having low social support were significantly and negatively associated with mental HRQoL scores. Finally, in Model 6, the interaction effects between numbers of unmet basic needs and age groups were tested. Several interaction effects were significantly and negatively associated with lower scores in physical and mental HRQoL. Compared to the two younger groups, in the oldest group, physical HRQoL scores fell by five and six points when there were 2- to 4-unmet needs (all, *p* < 0.05), while mental HRQoL scores fell by eight points when there were 4-unmet needs (*p* = 0.009).

### 3.5. The Trajectories of Physical and Mental Health-Related Quality of Life

The four-year trajectories of physical and mental HRQoL in persons living with HIV in the PSHP cohort were evaluated. Trajectories of physical and mental HRQoL scores illustrated the trend over time. 

We observed that mean scores were reduced by the number of unmet basic needs in physical HRQoL (Figure 3a,b). Similarly, mental HRQoL mean scores of participants were also reduced by the number of unmet basic needs. However, for both physical and mental HRQoL, the disparities and decrements in the oldest age group were the most pronounced when basic needs were unmet.

## 4. Discussion 

This is the first longitudinal Canadian study in persons living with HIV using individual growth curve modeling to investigate the rates of change in physical and mental HRQoL in relation to the varying number of unmet basic needs across age groups. 

Overall, we found a high number of unmet basic needs in access to food and clothing, inability to cover housing-related costs, and feeling at risk of homelessness in our PSHP sample, with approximately 86% of participants endorsing at least one of these items. The evaluation of changes in unmet basic needs over time was not significant, suggesting that these challenges were relatively stable over the four years. The oldest participants reported more ‘*met*’ needs and, gradually, fewer unmet needs, while the two younger participant groups (age 20–34 years old and 35–49 years old) reported fewer ‘*met*’ needs over time. 

There is evidence that older HIV adults endorse good health and enjoy life [53]. This might result from resilience, which reduces the negative influence of their daily life stress [54]. Adults aging with HIV in our cohort appeared to be stable with respect to unmet basic needs relative to the two younger age groups. While we cannot directly compare the prevalence of unmet basic needs in our cohort to other studies because of differences in the range and definition of unmet basic needs, the literature consistently reports a high prevalence of such challenges. One study concluded that HIV-positive men in Ontario, Canada, were nine times more likely to have unmet food needs than their HIV-negative male peers [34]. Hessol et al. observed that one in three older adults with HIV was food insecure [20]. Another recent study found that over half (58%) of their participants with HIV had unstable housing [55]. 

In our evaluation of HRQoL, we observed that both the physical and mental health scores of our participants fell below those of the Canadian general population [56], but that these scores slightly improved (less than one point each year) over the four years of our study. The overall rates of change, however, indicated that the observed within-individual variation contributes to only 5.0% (in physical HRQoL) and 7.0% (in mental HRQoL) of these overall changes. Given the receipt of combination antiretroviral therapy for 8.0 years, this small amount of change may reflect the characteristics of the original PSHP cohort, deliberately recruited because they faced numerous social challenges. This could have served as a barrier to rapid improvements in their health and well-being. 

The presence and severity of social determinants of health such as poverty [15] to income insecurity [19], lack of food or appropriate housing [20,21] or social support [24], loneliness [22,23], or psychological hardships such as stigma [19,23,24], depression [25], and substance abuse [25,26] are known to affect the health and well-being of this vulnerable HIV population. For older adults with HIV who face multiple social adversities, age-related illnesses constitute an added health challenge [25,27,28,29,30,31].

When we examined the effect of unmet basic needs on HRQOL, we found that the number of unmet basic needs were significantly and negatively associated with lower scores in both physical and mental HRQoL―and these relationships were more pronounced (controlling for selected covariates) in participants aged 50 and older. These findings have important clinical significance and implications for overall HIV care, particularly for older adults. Antiretroviral therapies alone cannot markedly improve HRQoL of people living with HIV [57,58]. Regular evaluations by medical personnel of basic needs (i.e., food, clothing, housing-related costs, and the subjective risk of homelessness) are strongly recommended. 

We found no interactions between *time* and the number of unmet basic needs, suggesting that the relationship between the risk of unmet basic needs and physical or mental HRQoL scores in this cohort was relatively stable. However, we did find that aging had a significant impact, as shown by the fact that HRQoL trajectories, both physical and mental, were significantly influenced by the number of unmet basic needs in the oldest aged group. These findings suggest that, as one ages, unmet basic needs become important determinants and predictors of health status in persons living with HIV. 

There are several potential limitations to this study. Our cohort was not initially designed to study unmet basic needs, and no validated instruments exist for their measurement. In addition, our data are based on self-reports for which there is no objective verification. Because of the nature of retro-observational prospective data, it is also self-evident that our results show only associations; they cannot prove causality. As well, about one of every six baseline participants was dropped from the analysis. We could not specify comorbidities because they were not listed in the PSHP files. However, we did control for many of the most common confounders. Our data were derived from a cohort recruited to address housing needs so that recruitment may have been skewed to a relatively disadvantaged population with more housing instability and increased unmet basic needs than would be expected. While some health information was incomplete, we were able to use a sophisticated statistical method, the individual growth curve, to manage missing data and to analyze the rates of change in our longitudinal data [59,60,61]. 

## 5. Conclusions

Our main findings are that basic needs (i.e., access to food security, clothing, and housing) can exert a significant and negative effect on both physical and mental HRQoL particularly as people living with HIV get older. Given these associations, a patient’s ability to meet basic needs stands out as an important indicator of overall health. Primary health care, specialty health care, case management, and care coordination are currently available for patients with HIV in most developed countries, but they have focused on care engagement, treatment adherence, viral suppression, attention to treatment side-effects, and longevity more than on well-being and quality of life. National governments have a major role to play to ensure their citizens’ well-being by passing legislation and providing funding and appropriate resources for interventions that benefit HIV populations. These include accessibility and timeliness of medical and psychosocial, patient-centered, evidence-based care and continuity. 

## Figures and Tables

**Figure 1 ijerph-18-12256-f001:**
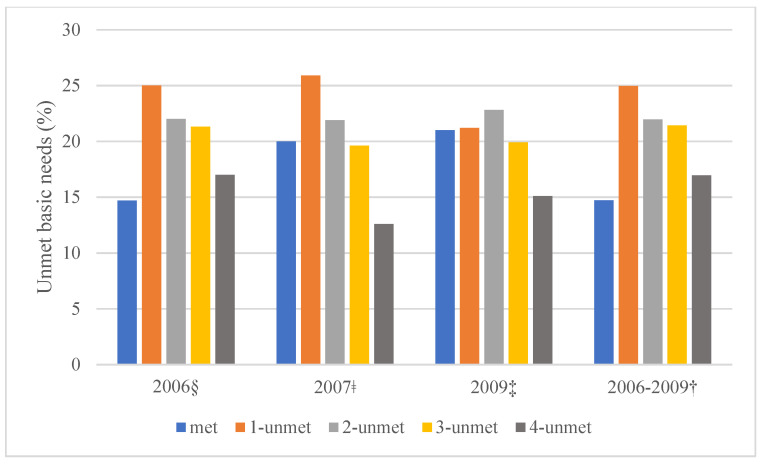
Distributions of unmet basic needs in people living with HIV at each time point (2006, 2007, 2009) and over time (2006–2009) in Positive Spaces, Healthy Places. § *p* = 0.001; ǂ *p* = 0.0001; ‡ *p* = 0.106; † *p* = 0.133.

**Figure 2 ijerph-18-12256-f002:**
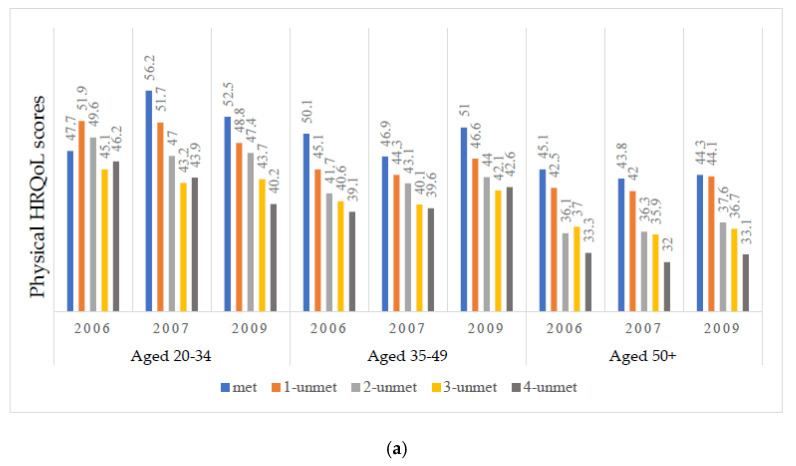
Unadjusted unmet basic needs on physical (**a**) and mental (**b**) HRQoL by age group at each time point in Positive Spaces, Healthy Places.

**Figure 3 ijerph-18-12256-f003:**
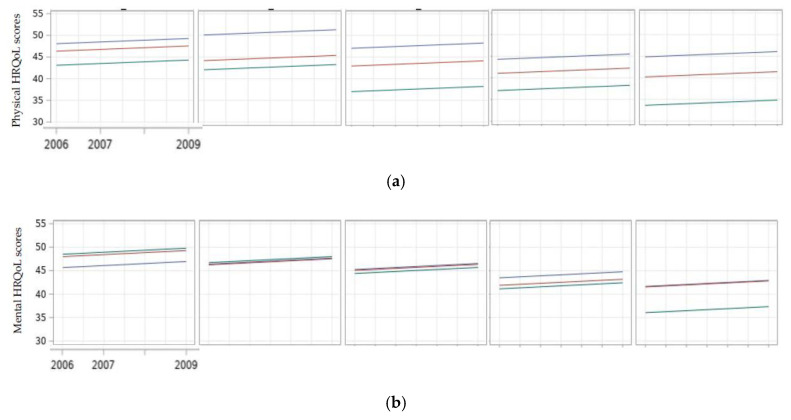
Trajectories of physical (**a**) and mental (**b**) HRQoL scores over time (2006–2009) in Positive Spaces, Healthy Places. Unmet basic needs (left to right) = met, 1-unmet, 2-unmet, 3-unmet, 4-unmet. Age groups: younger age (blue), middle-aged (red), older age (green).

**Table 1 ijerph-18-12256-t001:** Baseline sample characteristics (2006) in Positive Spaces, Healthy Places.

Variables	Total n = 538	Unmet Basic Needs n = 537					
		0-Unmet (Met) n = 79 (14.71)	1-Unmet n = 134 (24.95)	2-Unmet n = 118 (21.97)	3-Unmet n = 115 (21.42)	4-Unmet n = 91 (16.95)	*p* Value ǂ
	n (%)/ Median (IQR)	n (%)/ Median (IQR)	n (%)/ Median (IQR)	n (%)/ Median (IQR)	n (%)/ Median (IQR)	n (%)/ Median (IQR)	
Physical HRQoL summary ^‡^	42.36 (34.30, 51.66)	50.0 (39.58, 56.70)	47.26 (36.0, 54.88)	40.98 (33.28, 49.86)	40.26 (33.91, 46.89)	38.98 (29.69, 47.34)	<0.0001
Mental HRQoL summary ^‡^	43.39 (35.89, 53.42)	52.07 (42.22, 59.24)	50.25 (39.40, 55.90)	42.65 (34.68, 53.13)	41.73 (35.04, 49.64)	37.90 (30.94, 44.27)	<0.0001
Demographics							
Non-greater Toronto residents	213 (39.59)	19 (24.05)	49 (36.57)	49 (41.53)	56 (48.70)	40 (43.96)	0.002
Age (n = 507)							
Young, 20–34 ^†^	75 (14.79)	9 (12.16)	16 (12.40)	14 (12.73)	15 (13.64)	20 (24.10)	0.136
Middle-aged, 35–49	304 (59.96)	40 (54.05)	77 (59.69)	67 (60.91)	71 (64.55)	49 (59.04)	
Old, 50+	128 (25.25)	25 (33.78)	36 (27.91)	29 (26.36)	24 (21.82)	14 (16.87)	
Female	134 (24.91)	16 (20.25)	23 (17.16)	24 (20.34)	33 (28.70)	38 (41.76)	<0.0001
Caucasian	394 (73.23)	57 (72.15)	103 (76.87)	89 (75.42)	79 (68.70)	66 (72.53)	0.473
Heterosexual (n = 530)	195 (36.79)	25 (32.05)	30 (22.73)	36 (30.51)	53 (46.90)	50 (56.82)	<0.0001
SES/living and support conditions							
No high school degree (n = 536)	116 (21.64)	10 (12.66)	25 (18.80)	30 (25.42)	25 (21.74)	26 (28.89)	0.013
Low income < 1150 CAD/m (n = 515)	261 (50.68)	37 (50.00)	56 (43.75)	59 (53.64)	63 (55.75)	46 (51.11)	0.274
Unemployed (n = 537)	424 (78.96)	55 (69.62)	95 (70.90)	102 (87.18)	93 (80.87)	78 (85.71)	0.001
Living alone (n = 486)	259 (53.29)	41 (60.29)	66 (54.55)	63 (60.00)	51 (48.11)	38 (44.19)	0.026
Low social support < 43 (n = 525)	255 (48.57)	36 (45.57)	55 (41.67)	59 (51.75)	59 (53.64)	46 (51.69)	0.987
HIV status							
A history of AIDS diagnosis (n = 527)	271 (51.41)	35 (46.05)	74 (56.06)	61 (53.04)	56 (49.12)	45 (50.56)	0.888
CD4 T-cells < 200/mL (n = 454)	256 (56.39)	39 (54.93)	72 (64.29)	54 (51.92)	52 (53.06)	39 (57.35)	0.526
Health behavior risks							
Depressive symptoms (n = 481)	195 (40.54)	9 (13.24)	34 (28.57)	49 (45.79)	54 (52.94)	49 (57.65)	<0.0001
Drug abuse (n = 532)	102 (19.17)	7 (8.97)	25 (18.66)	26 (22.81)	23 (20.00)	21 (23.33)	0.036
Alcohol use disorder (n = 536)	90 (16.79)	9 (11.54)	19 (14.29)	26 (22.03)	23 (20.00)	12 (13.19)	0.444
A history of homelessness	213 (39.59)	27 (34.18)	47 (35.07)	50 (42.37)	52 (45.22)	36 (39.56)	0.158
A history of incarceration (n = 533)	158 (29.64)	18 (23.08)	30 (22.56)	45 (38.14)	39 (34.51)	26 (28.89)	0.085

^†^ Missing one person across levels of unmet basic needs. ^‡^ Mean (standard error) was applied for individual growth curve models. ǂ *p* value of Kruskal–Wallis, Cochran–Armitage, or Cochran Mantel–Haenszel tests, where appropriate. SES, social economic status.

**Table 2 ijerph-18-12256-t002:** Individual growth curve models for changes in Physical Health Summary (2006–2009) in Positive Spaces, Healthy Places (n = 1319).

	Model 1	Model 2	Model 3	Model 4	Model 5	Model 6
**Fixed effects**						
Intercept	43.30 (0.40) ^a^	42.69 (0.45) ^a^	46.19 (0.67) ^a^	50.98 (1.14) ^a^	54.63 (1.26) ^a^	52.50 (1.78) ^a^
Time		0.51 (0.15) ^b^	0.43 (0.15) ^b^	0.38 (0.15) ^c^	0.39 (0.16) ^c^	0.40 (0.16) ^c^
Unmet basic needs (met = ref.)						
1-unmet			−1.42 (0.67) ^c^	−1.47 (0.68) ^c^	−1.29 (0.70) ^d^	2.02 (1.87)
2-unmet			−4.09 (0.71) ^a^	−4.11 (0.72) ^a^	−3.75 (0.75) ^a^	−1.06 (1.96)
3-unmet			−5.63 (0.75) ^a^	−5.71 (0.76) ^a^	−5.30 (0.78) ^a^	−3.78 (1.91) ^c^
4-unmet			−6.69 (0.84) ^a^	−6.78 (0.85) ^a^	−6.40 (0.87) ^a^	−3.21 (2.15)
Age groups (20–34 = ref.)						
Middle-aged, 35–49				−4.17 (1.10) ^a^	−4.10 (1.07) ^b^	−1.72 (1.88)
Old, 50+				−8.85 (1.23) ^a^	−8.21 (1.22) ^a^	−4.98 (2.07) ^c^
Non-greater Toronto resident					−3.43 (0.76) ^a^	−3.34 (0.76) ^a^
Caucasian					−2.37 (0.84) ^b^	−2.61 (0.84) ^b^
Low social support < 43					−1.60 (0.60) ^b^	−1.68 (0.55) ^b^
Drug abuse ≥ 6 DAST scores					−1.67 (0.73) ^c^	−1.58 (0.73) ^c^
Interactions						
1-unmet × middle-aged						−4.21 (2.08) ^c^
1-unmet × old						−3.06 (2.29)
2-unmet × middle-aged						−2.38 (2.18)
2-unmet × old						−5.03 (2.48) ^c^
3-unmet × middle-aged						−1.52 (2.15)
3-unmet × old						−2.24 (2.49)
4-unmet × middle-aged						−2.91 (2.42)
4-unmet × old						−6.21 (2.83) ^c^
**Random variance**						
Intercepts (1,1)	70.27 ^a^	79.52 ^a^	71.19 ^a^	64.42 ^a^	57.53 ^a^	57.10 ^a^
Intercepts-slopes (2,1)	-	−4.32 ^c^	−3.97 ^c^	−3.78 ^c^	−3.46 ^c^	−3.32 ^d^
Linear slopes (2,2)	-	1.88 ^c^	1.56 ^c^	1.51 ^c^	1.59 ^c^	1.67 ^c^
Residuals	44.60 ^a^	39.93 ^a^	39.18 ^a^	39.01 ^a^	38.67 ^a^	37.30 ^a^
Observations used	1482	1482	1481	1395	1319	1319
Fit statistics and model tests						
-2LL	10,730.4	10,713.8	10,614.4	9943.4	9357.6	9321.9
AIC	10,736.4	10,721.8	10,622.4	9951.4	9365.6	9329.9
BIC	10,749.3	10,739.0	10,639.5	9968.3	9382.5	9346.8
Chi-square	508.42 ^a^	521.67 ^a^	470.48 ^a^	400.39 ^a^	328.23 ^a^	329.43 ^a^

Covariates: Sex, sexual preference, level of education, income, employment, living arrangement, CD4 T-cell, a history of AIDS, a history of homelessness, alcohol use disorder, and depression did not include due to omitting the significance of linear slopes or *time*, and/or not improving model fitness. *p* value: a < 0.001, b < 0.01, c < 0.05, and d = 0.07. -2 LL, -2 Log-likelihood ratio; AIC, Akaike Information Criterion; BIC, Bayesian Information Criterion.

**Table 3 ijerph-18-12256-t003:** Individual growth curve models for changes in Mental Health Summary (2006–2009) in Positive Spaces, Healthy Places (n = 1274).

	Model 1	Model 2	Model 3	Model 4	Model 5	Model 6
**Fixed effects**						
Intercept	45.14 (0.42) ^a^	44.20 (0.48) ^a^	48.61 (0.74) ^a^	48.56 (1.25) ^a^	55.03 (1.36) ^a^	52.48 (2.08) ^a^
Time		0.78 (0.18) ^a^	0.68 (0.18) ^a^	0.65 (0.18) ^b^	0.41 (0.20) ^c^	0.43 (0.20) ^c^
Unmet basic needs (met = ref.)						
1-unmet			−1.68 (0.78) ^c^	−1.65 (0.81) ^c^	−1.49 (0.80) ^d^	0.72 (2.23)
2-unmet			−4.41 (0.83) ^a^	−4.34 (0.85) ^a^	−2.98 (0.86) ^a^	−0.41 (2.41)
3-unmet			−7.20 (0.86) ^a^	−7.08 (0.89) ^a^	−5.89 (0.89) ^a^	−2.18 (2.32)
4-unmet			−9.60 (0.96) ^a^	−9.44 (0.99) ^a^	−7.39 (1.00) ^a^	−4.01 (2.51)
Age groups (20–34 = ref.)						
Middle-aged, 35–49				−0.40 (1.17)	−0.23 (1.12)	2.33 (2.84)
Old, 50+				−0.92 (1.32)	−1.00 (1.25)	2.84 (2.39)
Female					−1.97 (0.90) ^c^	−1.81 (0.90) ^c^
Low income < 1150 CAD/m					−1.31 (0.60) ^c^	−1.25 (0.61) ^c^
Unemployed					−2.88 (0.73) ^a^	−3.10 (0.74) ^a^
Low social support < 43					−5.23 (0.61) ^a^	−5.24 (0.61) ^a^
Drug abuse ≥ 6 DAST scores					−5.43 (0.80) ^a^	−5.32 (0.80) ^a^
Interactions (20–34 [ref.])						
1-unmet × middle-aged						−2.47 (2.46)
1-unmet × old						−2.52 (2.70)
2-unmet × middle-aged						−2.53 (2.63)
2-unmet × old						−3.70 (2.95)
3-unmet × middle-aged						−3.93 (2.57)
3-unmet × old						−5.19 (2.95) ^d^
4-unmet × middle-aged						−2.45 (2.79)
4-unmet × old						−8.44 (3.25) ^b^
**Random variance**						
Intercepts (1,1)	71.70 ^a^	79.72 ^a^	68.31 ^a^	67.38 ^a^	55.10 ^a^	55.20 ^a^
Intercepts-slopes (2,1)	-	−3.83	−4.67 ^c^	−3.83 ^c^	−5.64 ^c^	−5.67 ^c^
Linear slopes (2,2)	-	2.73 ^c^	2.32 ^c^	2.03 ^c^	3.70 ^b^	3.65 ^b^
Residuals	65.09 ^a^	57.70 ^a^	56.55 ^a^	57.40 ^a^	49.40 ^a^	49.31 ^a^
Observations used	1482	1482	1481	1395	1274	1274
Fit statistics and model tests						
-2 LL	11,140.5	11,117.4	10,982.7	10,348.7	9276.9	9240.9
AIC	11,146.5	11,125.4	10,990.7	10,356.7	9284.9	9248.9
BIC	11,159.3	11,142.6	11,007.8	10,373.6	9301.7	9265.8
Chi-square	354.41 ^a^	370.52 ^a^	296.42 ^a^	280.22 ^a^	189.23 ^a^	187.68 ^a^

Covariates: Ethnicity, sexual orientation, place of residents, level of education, living arrangement, CD4 T-cell, a history of AIDS, a history of homelessness, alcohol use disorder, and depression did not include due to omitting the significance of linear slopes or *time*, and/or not improving model fitness. *p* value: a < 0.001, b < 0.01, c < 0.05, d = 0.08. -2 LL, -2 Log-likelihood ratio; AIC, Akaike Information Criterion; BIC, Bayesian Information Criterion.

## Data Availability

De-identified raw data and materials described in the manuscript are freely available from the corresponding author on reasonable request.

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
