# Peer review of "Four-Year Trajectories of Health-Related Quality of Life in People Living with HIV: Impact of Unmet Basic Needs across Age Groups in Positive Spaces, Healthy Places"

_ijerph, 2021, doi:10.3390/ijerph182212256_

Round 1

Reviewer 1 Report

The manuscript titled “Four-year trajectories of health-related quality of life in people living with HIV: Impact of unmet basic needs across age groups in Positive Spaces, Healthy Places” examined the effect of unmet basic needs across age groups on changes in HRQoL over a 4-year period in persons with HIV and found low scores on physical and mental HRQoL were positively associated with higher number of unmet basic needs. This is a longitudinal study. The author observed and followed up for 4 years. Since there is no intervention, it's hard to directly study whether improve the unmet basic needs could help on the physical and mental HRQoL scores. Here are the major comments:

  1. Sample and data analyses:

Here, the author mentioned that “However, those who missed their follow-up appointment in 2007 were significantly more likely than others to report significant alcohol or drug use, history of incarceration or homelessness, and depression at their baseline interview. Those who were missing the 2009 interview were significantly more likely than others to be male, gay, bisexual or men who have sex with men, middle-aged, living alone and financially disadvantaged.” Since these covariates were significantly associated with the missed follow-up. The missing can not be considered as randomly missing. The author shall adjust these covariates in the model.

  1. Table 1, N or n, not consistent.
  2. Figure 3, is this the trend analysis? P-value is not reported.

Author Response

Our responses for the Reviewer 1

1) Since there is no intervention, it's hard to directly study whether improve the unmet basic needs could help on the physical and mental HRQoL scores.

We agree with the comment. We address this issue in Discussion Line 360-1, new revision.

2) Missing data, the missing cannot be considered as randomly missing. The author shall adjust these covariates in the model.

Type of missing data and methods of dealing missing data have been discussed in a wonderful article by J.T. Newsom, http://web.pdx.edu/~newsomj/semclass/ho_missing.pdf.

In our longitudinal study, missing data were not merely involved in missing data but missing follow up (i.e., study participants) during our study follow up in 2007 and 2009. In our knowledge, missing follow up happened to any longitudinal studies. This type of missing (missing follow-up) might not allow researchers to use a standard method (i.e., imputation) to resolve missing data.

Fortunately, we employed the very sophisticated modelling method of “growth curve models” to evaluate and assess the rate of changes of physical and mental HRQoL associated with unmet basic needs across age groups in our HIV cohort, Positive Spaces, and Healthy Places, Toronto, Canada. This sophisticated model incorporated with the mixed-effects regression method.

As per ref. 50 and 51, mixed-effects regressions can handle missing data and irregular time measures (i.e., unbalanced designs) for both time-invariant and time-varying variables. Please note, in our method section (Line 155 to 160), we explain how we handled the missing data.

We also employed variables that did not affect our models. Covariates were added while maintaining the significance level of intercepts and linear slopes, as well as time (per ref. 51). For these reasons, we feel that there are no reasons to re-test our models. 

For some covariates that affected our models or did not improve our models, we indicated them in the footnote of Table 2 and Table 3.

3) N or n in Table 1:

We now use ‘n’.

4) Figure 3:

Figure 3 (a/b) for physical and mental HRQoL did not come from a “trend analysis”. We produced the graphs from SAS analyses to show the trajectory of physical HRQoL and mental HRQoL. SAS does not give p-values for these specific graphs.

Please note, Figure 3 (i.e., the trajectory of physical HRQoL and mental HRQoL) are exactly the same results in Table 2 and Table 3, showing interactions between “unmet basic needs x age groups”. We have included a p-value and a standard error (SE) for each interaction in Table 2 and Table 3.

We have now submitted a revision of Figure 3 (a/b). We believe the new Figure 3 allows readers to better understand the trajectories of physical and mental HRQoL in our study population.

Reviewer 2 Report

The advancement of combined antiretroviral therapy (cART) has dramatically improved the outlook of human immunodeficiency virus type 1 (HIV-1) patients, and therefore it is now considered a chronic disease. The authors point out an important caveat: antiretroviral therapies alone are not enough to improve the quality of life of people living with HIV. The article gives a good overview of different groups of people with HIV and how the unmet needs negatively affect the HRQoL. Interestingly, older HIV adults appear to be more stable and enjoy life more than younger-aged groups. It is also interesting that there was no significant difference in time and number of basic unmet needs. Even with many limitations, this study highlights an important issue that people living with HIV face today, even with such advancements in the treatment, which are very important to be addressed.

Minor comments:

The figure 2 bar graphs are not legible, and I would suggest redoing them like the figure 1 bar graph.

In the discussion, it would be good to get authors prospective on how society and/or government entities can help people living with HIV to decrease the number of basic unmet needs and improve their quality of life.

Author Response

Our responses to the Reviewer 2

1) Figure 2:

We have now submitted a revision of Figure 2 (a/b). We also include data points in the new Figure 2 (a/b), which allows readers to easily see the relationships of physical and mental HRQoL score, and unmet basic needs by age groups at each study point.

We also made a slight revision in the color bar for “4-unmet”, consistent with Figure 2 (a/b).  

2) In the discussion, it would be good to get authors prospective on how society and/or government entities can help people living with HIV to decrease the number of basic unmet needs and improve their quality of life:

Thank you for this suggestion, which we have now put in the Conclusion (lines 370-81)―Our main findings are that basic needs (i.e., access to food security, clothing, and housing) can exert a significant and negative effect on both physical and mental HRQoL particularly as people living with HIV get older. Given these associations, a patient’s ability to meet basic needs stands out as an important indicator of overall health. Primary health care, specialty health care, case management, and care coordination are currently available for patients with HIV in most developed countries, but they have focused on care engagement, treatment adherence, viral sup-pression, attention to treatment side-effects, and longevity more than on well-being and quality of life. National governments have a major role to play to ensure their citizens’ well-being by passing legislation and providing funding and appropriate resources for interventions that benefit HIV populations. These include accessibility and timeliness of medical and psychosocial patient-centered, evidence-based care and continuity.

Reviewer 3 Report

This is an excellent piece of work. The authors provide literature relevant to the topic in the introduction, which is excellent. The methods and results are clearly presented. I believe it still requires minor editing. The sentence on lines 306 to 307 in the "Discussion" section is a little confusing. Overall, I believe the paper reports findings that are relevant and may have implications for research, programs, and possibly policy for people living with HIV not only in Canada, but also in other areas or regions where HIV prevalence remains high.

Author Response

Our response to The Reviewer 3

1) Lines 306 – 307: In older adults with HIV, both age-related illnesses and social determinants of health constitute health challenges (ref. 25, 27-31).

We re-wrote this (lines 339-40): For older adults with HIV who face multiple social adversities, age-related illnesses constitute an added health challenge (ref. 25, 27-31).

Round 2

Reviewer 1 Report

I have no further comments on this study.